# Nursing Experience of New Nurses Caring for COVID-19 Patients in Military Hospitals: A Qualitative Study

**DOI:** 10.3390/healthcare10040744

**Published:** 2022-04-16

**Authors:** Young-Hoon Kwon, Hye-Ju Han, Eunyoung Park

**Affiliations:** 1College of Nursing, Chungnam National University, Daejeon 35015, Korea; hoon8238@naver.com; 2Nursing Department, Armed Forces Capital Hospital, Seongnam 13574, Korea; fns0015@naver.com

**Keywords:** COVID-19, military hospitals, nurses, qualitative research

## Abstract

This qualitative study explored the experiences of new nurses with less than one year of clinical experience in caring for COVID-19 patients in a military hospital. In-depth interviews were conducted with six new nurses working in a negative-pressure isolation unit of the Armed Forces Capital Hospital. Data were analyzed using the phenomenological method proposed by Colaizzi, and 12 themes were derived and classified into four clusters: burden of nursing in isolation units; hardship of nursing critically ill patients; efforts to perform nursing tasks; positive changes through patient care. The participants were anxious while caring for COVID-19 patients with severe illness due to a lack of clinical experience. Furthermore, the wearing of heavy personal protective equipment impeded communication with patients, leading to physical and psychological exhaustion. However, they tried to utilize their own know-how and provide the best nursing care, resulting in them gaining confidence. Participants were able to think critically and took pride in being military nursing professionals. This study is meaningful as it provides insight into the experiences of new military nurses who were rapidly dispatched during a national medical crisis. The results can be applied to develop future strategies aimed at improving new nurses’ competency in military hospitals.

## 1. Introduction

The coronavirus disease 2019 (COVID-19) is a respiratory syndrome caused by the SARS-CoV-2 infection and is associated with rapid rates of infection and mutation. The World Health Organization (WHO) declared the outbreak of a global pandemic on 11 March 2020, less than three months after the first confirmed case of infection in China was reported [1]. Despite various efforts to develop therapeutics and preventive vaccines worldwide, the COVID-19 pandemic has continued to spread, with the number of infections fluctuating globally [2]. Approximately 400 million confirmed cases have been reported worldwide [3], and there is currently no clear line of treatment for COVID-19 infections [4].

In South Korea, the first confirmed case of COVID-19 was reported on 20 January 2020. A month later, mass outbreaks were reported in several areas, and COVID-19 subsequently spread rapidly nationwide [5]. The Centers for Disease Control and Prevention of the Ministry of Health and Welfare established the Central Countermeasures Headquarters. On 23 February 2020, the infectious disease crisis alert level changed from caution to severe [5]. Additionally, an active system for strengthening patient monitoring, providing diagnostic tests for suspected cases, and managing patients was implemented to prevent transmission of COVID-19 within the community [6]. The South Korean government had already strengthened the infectious disease management system, including expanding negative pressure isolation facilities, as part of follow-up measures in the aftermath of the mass infection and spread of the Middle East Respiratory Syndrome (MERS) in 2015 that resulted in numerous deaths.

Despite such efforts, patients were still vulnerable to respiratory infectious diseases such as COVID-19 [6]. The medical system that focused on the initial response has faced a shortage of basic personal protective equipment, systems for diagnostic tests, beds in the intensive care unit, and medical personnel labor. This has led to burnout among medical staff [7]. As a result, the Ministry of National Defense has allotted several military hospitals for COVID-19 treatment to meet the demand for hospital beds, and dispatched large-scale military medical personnel to private hospitals for active early response to the COVID-19 pandemic [5]. As COVID-19 is an infectious respiratory disease, prior education for wearing personal protective equipment and caring for patients is fundamental for successful treatment [8]. However, owing to the urgent dispatch of medical staff due to the current circumstances, the short training and preparation time was supplemented with individual clinical experience. Among the dispatched military medical personnel were new nurses who had insufficient clinical experience and no experience in providing care for infectious disease patients [9].

New nurses are those who possess a nurse’s license and have less than 12 months of clinical experience [10]. New nurses are exposed to various stress factors in clinical practice due to insufficient knowledge and experience, and face difficulties in communicating with patients, family caregivers, and other staff. For these reasons, in the first year of adapting to the clinical nursing environment, new nurses often consider leaving the workforce [11]. Additionally, new nurses lack self-confidence in performing new tasks [11] and show increased rates of shock when they are assigned to unwanted departments or exposed to poor working conditions [12]. This shock causes confusion stemming from the disparity between the values they pursue at work and those they learned at school, causing new nurses to become overwhelmed by their workload and experience disharmony in their personal lives [13,14]. As a result, new nurses misadjust to the work environment, which interferes with their job immersion and leads to burnout [15]. Thus, new nurses require a lot of physical consideration and guaranteed conditions before they adapt to unfamiliar work environments and tasks. Based on these characteristics, it is expected that new nurses likely experienced many difficulties in caring for COVID-19 patients in military hospitals.

Caring for COVID-19 patients is accompanied by various challenges. In studies on medical personnel caring for COVID-19 patients, the staff complained of mental health problems such as depression, anxiety, and insomnia [16], and experienced physical and psychological problems such as helplessness, hopelessness, and anger due to the lack of adequate resources and excessive work [17]. In particular, nurses who provided primary care for patients during pandemics experienced high psychological distress [18]. As the COVID-19 pandemic continues, the body of literature on medical personnel caring for COVID-19 patients is growing. However, few studies have evaluated new nurses, who likely face greater difficulties from nursing infectious disease patients and insufficient clinical experience [19]. Moreover, although military nurses are obliged to perform duties immediately in national crises, the conservative nature of the military has prevented studies on new nurses in military hospitals. In particular, no study has evaluated the experience of caring for COVID-19 patients in special environments, such as negative pressure facilities.

Therefore, this study used a descriptive, phenomenological approach to investigate the lived experiences of new nurses in military hospitals, who were caring for COVID-19 patients in negative pressure nationally designated isolation units (NDIUs). This study aimed to understand in depth the essential structures of new nurses’ vivid experiences, and provide basic data to establish strategies for strengthening military nursing capabilities in a medical environment that is rapidly changing because of various infectious diseases.

## 2. Materials and Methods

### 2.1. Study Design

This qualitative study used a descriptive, phenomenological approach [20] to explore the experiences of new nurses caring for COVID-19 patients in NDIUs.

### 2.2. Participants and Setting

Purposive sampling was used to select participants who could best express the phenomena related to new nurses’ experiences. The participants of this study were new nurses with less than one year of clinical experience, who had obtained a nurse’s license in 2020, were assigned to military hospitals nationwide, and had experiences in nursing COVID-19 patients in NDIUs. Among 12 new nurses who were assigned or dispatched to NDIUs at the Armed Forces Capital Hospital in Seongnam, South Korea, eight nurses expressed the intention to participate in this study. During the study, two new nurses returned to their unit and refused to participate. Therefore, six new nurses were finally included in the analysis.

The participants worked in negative pressure units with more than 1000 inpatient beds in the Armed Forces Capital Hospital. The Armed Forces Capital Hospital serves the Armed Forces, discharged soldiers, and their families. Negative pressure units are NDIUs and can treat 8 critically ill and 40 mildly ill patients simultaneously. In addition, negative pressure units admit patients with the highest priority in cases of new infectious diseases.

### 2.3. Data Collection Methods and Procedures

Data were collected through one-on-one, in-depth individual interviews with the six participants based on the methods suggested by Coyne [21] and Kuzel [22], between 28 January and 23 August 2021. Due to the COVID-19 pandemic, data were collected over the phone.

All interviews were conducted by an interviewer, a university professor who is an experienced qualitative researcher and does not have any personal or professional relationships with any participants to understand their natural responses. Each participant was interviewed two to three times until no new concepts about the phenomena could be identified. Each interview lasted 40 to 65 min. The telephone interviews were conducted in a quiet and comfortable seminar room in the hospital to allow participants to openly express their thoughts and feelings without being disturbed. During the interviews, the participants were asked to talk freely about their thoughts, opinions, and experiences in response to the interviewer’s questions. The interviews were recorded and the initial interviews were conducted across approximately four weeks on items related to major phenomena. The second and third interviews consisted of additional questions and were each conducted across a four-week period. The recorded contents were transcribed by the first author, and the transcript was reviewed by each of the participants to ensure the accuracy of the contents. The interviews were conducted in Korean, and the contents of the interviews were translated into English by two professional translators and reviewed repeatedly by researchers for publication.

The interview questions started with broad and open semi-structured questions such as “what is your experience with caring for COVID-19 patients in NDIUs?” and were thematized according to the responses of the participants. Table 1 shows examples of the key interview questions.

### 2.4. Data Analysis

We employed Colaizzi’s phenomenological method—a descriptive phenomenology based on Husserl’s philosophy—which focuses on an exhaustive description of essential structures of the phenomena as well as unique experiences of the participants [20]. Data were analyzed using the phenomenological method used by Colaizzi to obtain an exhaustive description of the phenomena, as well as to uncover the overall structure of experience from themes generated by participants [20]. In step 1, each researcher read the interview transcript repeatedly to grasp the meaning related to the research phenomenon from the original data and to understand the participants’ experiences. Parts of the interview that were considered meaningful were underlined. In step 2, each researcher compared the underlined parts and read the manuscripts repeatedly to find phrases and sentences that directly related to the experiences of the participants. Parts that were expressed differently but had similar contents were derived as unified statements by collecting the opinions of the researchers and were recorded by the first author. In step 3, the important contents were re-stated in general form, and two researchers read the manuscript to find significant meanings. In step 4, each statement and the significant meanings identified were analyzed in detail through a discussion among the researchers. A total of 12 themes and 4 theme clusters were derived and categorized (Table 2). In step 5, the derived themes and theme clusters were described as the participants’ experiences. In step 6, the first author completed phenomenological writing according to the overall flow and revised the writing through discussion and consensus with the co-researchers. In step 7, to confirm the validity of the essential structure of the experience of COVID-19 nursing patients at military hospitals, the results were shown to two participants. The participants assessed whether the results were consistent with their experiences.

### 2.5. Ethical Considerations

This study was approved by the Institutional Review Board of the military hospital where this study was conducted (AFMC-202101-HR-002-01). The first author held a senior position in the same institution as the participants, which risked limiting their voluntary participation; hence, the corresponding author who did not work in the same institution and had no conflicting interests explained the study purpose and procedure, the participants’ right to voluntarily participate and withdraw from the study, and the guarantee of anonymity and confidentiality through individual phone consultations, in order to confirm the intention for participation. The participants provided their written informed consent before the interviews were conducted. In addition, the first author, who might have been responsible for evaluating the participants, did not conduct the interview to prevent them from responding differently from their original experience due to their concerns about the evaluation. During the study period, all staff performance appraisals in the workplace were also delegated to others not related to the research team.

### 2.6. Research Rigor

The criteria used by Lincoln and Guba [23] were applied to ensure the rigor of this study. First, the truth values obtained were shown to be realistic, confirming the credibility of the study results. The results were shown to two participants to confirm consistency with their experiences. Second, auditability was achieved by storing the transcript, analysis process, and results, following the data analysis procedure used by Colaizzi [20], and with the results reviewed by a nursing professor. Third, applicability was achieved via the continued collection of data until no new data were revealed in the interview contents of the participants. Fourth, neutrality was achieved by excluding bias. The first author with the same affiliation as the participants was excluded from the process of participant selection and data collection. A corresponding author with extensive experience in qualitative research conducted the interviews.

## 3. Results

Six participants were finally included in the analysis. The participants had clinical experiences of 6 to 11 months in military hospitals. One participant had one month of experience, while five participants had between 6 and 11 months of experience in the NDIUs, respectively. The participants comprised three women and three men. All participants were second lieutenants aged 23–24 years and were non-religious.

A total of 12 themes and four theme clusters were derived from the experiences of the new nurses caring for COVID-19 patients in NDIUs of military hospitals (Table 3). The theme clusters were as follows:Burden of nursing in isolation units;Hardship of nursing critically ill patients;Efforts to perform nursing tasks;Positive changes through patient care.

### 3.1. Burden of Nursing in Isolation Units

The participants expressed their confusion when they were first assigned to NDIUs. The participants were afraid of caring for isolated patients in the negative pressure unit, without sufficient nursing practice in caring for infectious disease patients. Above all, the participants were worried about their interactions with patients in the isolation units.

#### 3.1.1. Fear of a New Circumstance

As the participants were assigned to the isolation units, they were confused by their lack of knowledge and clinical experience in caring for COVID-19 patients. They were worried about being infected and potentially transmitting COVID-19 to colleagues or family members. However, they were also upset by the prevailing prejudice that they had a higher risk of transmitting COVID-19 to others.


*“I thought it may be too much for new nurses to care for COVID-19 patients while wearing a powered air-purifying respirator (PAPR) without being familiar with basic skills. I was worried that I may be infected while caring for COVID-19 patients and that I may transmit the infection to my family and others around me.” (Participant 1)*



*“I was upset to see the patients thinking that the nurses have a high risk of transmitting COVID-19 and maintaining distance from us.” (Participant 2)*


#### 3.1.2. Communication Difficulties with Isolated Patients

Participants experienced difficulty in communicating with patients as they could not talk to them face-to-face and were forced to rely on intercoms. In addition, the participants’ ability in responding to small requests by patients was limited, as they were required to perform numerous nursing tasks within a short amount of time while wearing personal protective equipment in the isolation rooms. The sound of surrounding medical equipment and their own personal protective equipment caused difficulties in hearing the voices of patients and caring for them and their families.


*“When I failed to draw blood from a patient, the patient became very emotional. I could not see the patient’s face, so interaction was difficult.” (Participant 2)*



*“An elderly patient felt frustrated about the negative pressure facility itself. No matter how many times I explained, the patient tried to open the automatic door with force and broke the doors. Some family members asked me to deliver food to the patients. It was difficult.” (Participant 4)*


### 3.2. Hardship of Nursing Critically Ill Patients

As most COVID-19 patients in the unit were severely ill, caring for them amid a staff shortage exhausted the participants physically and psychologically. The participants experienced a loss of confidence and were limited by accumulated fatigue, meaning they could not be of help to the patients and their colleagues.

#### 3.2.1. Nervous about Caring for Unfamiliar, Critically Ill Patients

The participants were confused and nervous about caring for severely ill patients and older patients with various underlying diseases, who were different from the usual active-duty soldiers these nurses were accustomed to caring for in military hospitals. In particular, participants experienced difficulties in nursing COVID-19 patients who had a lower level of self-care than soldiers.


*“The patients were not young patients in their 20s and 30s that I usually care for in the military hospital. I was confused about the various underlying diseases and drugs for elderly patients.” (Participant 5)*



*“The patients were not capable of independently caring for themselves. So, we had to do everything, from basic personal hygiene management to L-tube feeding, cleaning urine and feces, and changing their position in bed.” (Participant 4)*


#### 3.2.2. Physically Exhausted

The participants wore heavy personal protective equipment and were drenched in sweat from the accumulation of heat and moisture. They were exhausted from staying in the isolation rooms and caring for the patients for more than three hours while facing breathing difficulties themselves. Accumulated fatigue led to physical exhaustion and caused the participants to become more sensitive.


*“The gloves and protective suit really gave me a lot of difficulties as a new nurse. Fatigue and exhaustion were severe. The work was vigorous, and I became tired and more sensitive.” (Participant 1)*



*“When I wore PAPR, it was too hot, hard to breathe, and drained my stamina quickly.” (Participant 3)*


However, the new nurses ensured that they catered to basic patient needs such as food and water intake while wearing the protective gear. This, along with the lack of labor, led to an overload of work for both new and experienced nurses.


*“When you wear a protective suit, you face a lot of difficulties in making movements. If you are in a hurry to go to the bathroom or if you want to drink water, you have no choice but to endure it. The suit was also not ventilated, so when I got out of the hospital room and changed clothes, the operating clothes I wore inside the protective suit were wet from sweat.” (Participant 2)*


#### 3.2.3. Psychological Withdrawal

Participants expressed that applying their learnings from undergraduate studies in clinical practice was the most difficult part of their experience. As new nurses, the participants worried that their inexperienced nursing skills may harm the patients. Furthermore, they felt compassion for their colleagues, as the participants believed that they were not doing their share of tasks as efficiently as other members of the nursing team in the isolation units. On being criticized for their mistakes, their self-esteem was hurt. The participants experienced a lack of self-confidence and felt a sense of self-doubt.


*“When I thought I really could not do the given task, I sometimes asked a senior nurse outside (the isolation room). I felt ashamed and sorry that I could not even see their faces. (In the isolation room) I experienced great pressure as I had to perform the tasks alone. I wanted to do everything perfectly, the way the patients wanted. I was disappointed that I was not able to do so, and I lost confidence.” (Participant 2)*


Moreover, the participants were worried that they could not cope effectively with their lack of clinical experience in emergencies. The participants were not confident in their simple nursing skills and were psychologically withdrawn when confronted with high-level nursing tasks that involved medical equipment, such as a ventilator or high-flow oxygen supply.


*“When a patient became suddenly ill, I panicked. I was the only one caring for the patient (in the isolation room). I was afraid. I was anxious that ‘what if I make a mistake and harm the patient?’ If I made a mistake, I felt like I was being a burden. I felt confused. This psychological burden was heavy on me.” (Participant 1)*


### 3.3. Efforts to Perform Nursing Tasks

Although the participants were physically and psychologically exhausted, they aimed to do their best in providing nursing care for COVID-19 patients, which was their first task as a nurse in the military hospital. Thus, the participants explored their own ideas and self-study to comply with patient safety rules and work efficiently. Participants actively helped and encouraged their colleagues and coped with stress through teamwork.

#### 3.3.1. Studying Hard to Provide Skilled Nursing

To effectively perform the assigned tasks, participants believed that they needed to study further. The theories learned in school and clinical practice differed greatly, and they believed that they needed to learn new skills and practices. The participants self-studied, practiced nursing skills in their spare time, and sometimes actively sought help from experienced nurses to advance their abilities.


*“I watched videos on core skills, reviewed the skills on my own, and went to work. Changing the position of the patients on the bed is supposedly a simple task. However, I have only tried the task using a model in school. I had no practical experience and had to review the method on my own. I realized that studying from a book and actually caring for patients as a nurse were completely different.” (Participant 2)*



*“I asked for feedback from my seniors on what I was studying. If there was something I did not know regarding the medical records of patients, I studied the topic in my spare time after work. I lacked some skills and techniques, so I tried to make a plan to apply the skills in an easier way.” (Participant 3)*


#### 3.3.2. Searching for Own Know-How

The participants began to identify their own methods to increase the efficiency of nursing work through trial and error. Prior to entering the isolated units, where access was not free, participants mentally simulated the movements in the units and work order timings. Participants found that taking notes of necessary tasks was also helpful.


*“I think it’s helpful to go into the isolation room and simulate in your head which room to enter and which skill to perform. In a general ward, if you did not bring a tourniquet for a lab test, you can simply go to the station and bring a tourniquet. But in isolation rooms, you have to tell colleagues outside the room to deliver the items. This takes twice the time and gives additional tasks to the colleagues outside the room. So I think it is important to minimize such things.” (Participant 4)*



*“I can’t remember many things. If I rely on my memory, there is a risk of safety accidents. So, I wrote down the things to be done on a post-it note before entering the hospital and stuck the note on my person.” (Participant 5)*


#### 3.3.3. Showing Comradeship and Encouraging Each Other

The participants were new nurses and beginner officers. They attempted to overcome stress by encouraging each other despite having an exhausted body and lonely mind, while obeying military regulations, such as restrictions on going out due to the COVID-19 pandemic. Although there is a clear rank hierarchy in the military, encouraging each other through friendship, sharing stories, and empathizing with each other helped the participants overcome the difficulties of their work.


*“I think the most difficult time was when I felt lonely. But there were a lot of people around me who I could understand and empathize with. I felt relieved when I talked to them on the phone.” (Participant 2)*



*“We were of different ranks. But they were my senior and junior colleagues. We shared sorrows and joys, and our bonds became much stronger.” (Participant 3)*



*“I had fun working thanks to my colleagues. Sometimes it was not my turn to go into the isolation rooms, but I would go and help with work, and I also received some help as well.” (Participant 4)*


### 3.4. Positive Changes through Patient Care

Participants experienced various changes as they cared for COVID-19 patients in NDIUs. As they adapted to performing nursing work for isolated patients, the participants gained confidence and grew as individuals; furthermore, they tried to care for and be considerate of the patients. The participants were able to think progressively and critically regarding the management of infectious disease patients and took pride in their status as military nurses while witnessing the recovery of patients.

#### 3.4.1. Gaining Confidence

As the nurses’ experience in caring for COVID-19 patients increased, they were able to effectively adapt to the demands of their role and prioritize tasks, thereby gaining confidence. The participants were able to focus on the patients with a more relaxed mind as they handled tasks quicker, and thereby felt proud of themselves.


*“Caring for COVID-19 patients is different from working in the general wards. (omitted). When I see myself solving problems without major difficulties or issues during busy times, I feel that I have adapted to the work.” (Participant 1)*



*“It was when I had the time to do things efficiently and quickly and listen to the patients. I was always busy with the assigned tasks, and I could not afford to look back at the patients. But now I can look around the rooms once more and check on patients more often. I can say that ‘I have adapted to the work.’” (Participant 6)*


#### 3.4.2. Striving for Patient-Centered Care

The participants began to feel a sense of duty in their roles as nurses in the isolation unit. They developed compassion, empathy, and understanding for the patients. The participants became independent and capable nurses by prioritizing the needs and experiences of patients.


*“While I was taking care of a patient, the patient’s condition deteriorated, and the patient was transferred to a higher-level hospital. Maybe I grew as a nurse. The experience was heartbreaking and difficult. I was scared. I was sorry to see the patient sick and being transferred. I felt a sense of duty to do better.” (Participant 1)*



*“When the patients got angry and complained, I thought ‘it must be difficult for them’, ‘it must hurt their mind as they cannot breathe properly’. Even though these may mean the same, I ask ‘Are you alright? What is most uncomfortable for you right now?’ If the patient answers ‘nothing is uncomfortable.’ I tried to understand the patient by asking ‘how are you feeling today.’” (Participant 6)*


#### 3.4.3. Thinking Critically

The participants, who were initially overwhelmed with their assigned tasks as new nurses, became aware of the need for efficient workforce management in providing nursing care for COVID-19 patients, and to minimize the risk of patient safety accidents. Furthermore, participants broadened their perspectives to critically think about the lack of practical national institutional support.


*“I think that the experience of working with experienced medical personnel to nurse COVID-19 patients has helped to improve the level of care and nursing capacity of military hospitals. Despite the lack of labor and requests for support, we never received quicker feedback or political, physical, and financial support. They only ‘thanked’ us in words, which felt foreign.” (Participant 1)*



*“I think that if all acting nurses have to be in the isolation room, there must be a system that can help the nurses with administrative tasks outside the isolation room.” (Participant 2)*


#### 3.4.4. Feeling Proud as a Military Nursing Officer

The participants felt rewarded and happy that they were doing their part on the frontlines in caring for COVID-19 patients. The participants felt proud of protecting and serving the people during a national crisis as both soldiers as well as nurses who take care of patients.


*“I felt proud, rewarded, and happy hearing messages like ‘thanks to you I recovered speedily.’ Looking back at it later, I think it will be a very happy moment in my life.” (Participant 2)*



*“The whole world is in the middle of the COVID-19 pandemic. It will be rewarding as a nursing officer to have worked at the forefront as a soldier and a nurse in a place that provides the highest level of medical service in the military.” (Participant 4)*


## 4. Discussion

This study was conducted to explore and understand the essential structures of new nurses’ experience of nursing COVID-19 patients at military hospitals, using a phenomenological approach. A total of 12 themes and 4 theme clusters were derived. The discussion of the phenomenon observed in the study, focusing on the clusters of themes, is as follows.

First, as described in the theme cluster “burden of nursing in isolation units,” new nurses at military hospitals were confused about being assigned to caring for COVID-19 patients. New nurses in private hospitals caring for COVID-19 patients also reported similar experiences [24]. The assignment of units is a factor that has the most sensitive effects on the initial clinical adaptation of new nurses [25,26]. The task of nursing infectious disease patients in a negative-pressure isolation unit is not easy, not only for new nurses but also for senior nurses with abundant clinical experience. In 2015, the nurses who cared for MERS patients in South Korea also expressed a similar fear of being exposed to MERS, and worried that they could spread the virus to colleagues and family members [27]. Additionally, considering the lack of educational content on quarantine environments and practical experience in nursing infectious disease patients in the current nursing curriculum, new nurses likely experienced greater anxiety and confusion than experienced nurses. Recently, practical environments using augmented or virtual reality have shown significant practical educational effects. Therefore, to improve the clinical skills and self-efficacy of new nurses, it is important to establish a practical system in the nursing curriculum that combines science and technology with systematic education for special nursing, such as caring for infectious disease patients [28,29].

The participants also reported that they experienced difficulties in communicating with patients. COVID-19 patients felt anxious about being quarantined and isolated from their families and society [30,31,32] and did not comply with the treatment guidelines. In such uncertain situations, large conflicts and lack of resources only increase burnout in nurses [33]. Therefore, effective therapeutic communication education for nursing students and nurses must be strengthened to improve communication with isolated infectious disease patients and to alleviate conflicts.

Second, in the theme cluster “hardship of nursing critically ill patients,” new nurses were nervous about rapidly worsening symptoms in older patients with severe illness and various unfamiliar underlying conditions, unlike general military hospital inpatients. As a result, the participants experienced physical and psychological burnout. New nurses who start their careers as clinical nurses in special departments such as the intensive care units (ICUs) and emergency rooms perceive these departments as harsh environments, and often experience a strong shock [34]. However, due to the personnel structure and work specificity in the military medical system, new nurses cannot be completely excluded from working in special departments. The same circumstances are observed in the general medical system devoted to responding to the COVID-19 pandemic. Therefore, in both general and military medical systems, it is inevitable that new nurses are assigned to respond to infectious diseases to supplement the shortage of medical personnel [19]. Therefore, when new nurses are inevitably assigned to special departments, a system must be established to guarantee sufficient rest and safe working conditions, such as rotation of workplace duties and allocation of adequate labor to prevent exhaustion among medical staff. Moreover, considering that new nurses in military hospitals may need to treat various groups such as older individuals, children, and pregnant women in times of national crisis, active academic exchange between the civil and military medical systems must be prepared to ensure nurses in military hospitals gain diverse clinical experience through caring for and interacting with different groups.

New nurses were psychologically withdrawn as they made repeated mistakes due to their inexperienced nursing skills. New nurses show a lower level of resilience than experienced nurses [35,36], and the lack of clinical experience is associated with poor ability to cope with stress, which in turn leads to higher psychosocial stress [11,37]. In particular, for groups in which senior nurses criticized junior nurses, the shock scores of new nurses were higher than that for groups in which junior nurses were not criticized [38]. Thus, the psychological difficulties, such as a lack of self-confidence, of new nurses caring for COVID-19 patients in the hierarchical military culture and constricted space of isolation units may be another stress factor. Communication and mutual respect between new and experienced nurses are fundamental to successful nursing. In particular, in sharing and deciding on new treatment policies during the COVID-19 pandemic, clear and timely communication between healthcare professionals is necessary [39].

Third, as shown in the theme cluster “efforts to perform nursing tasks,” new nurses gradually made efforts to accumulate professional knowledge and practice their skills through trial and error. Nursing patients with new infectious diseases require nursing knowledge, which is the basis of an integrated perspective [27]. In order for new nurses to develop into professional nurses, they must not only acquire basic nursing skills, but also develop an integrated ability to solve problems in rapidly changing health statuses in patients. Such skills can be acquired through effective clinical practice education and evaluation. Therefore, it is necessary to develop practice courses and appropriate evaluation tools for special nursing by nursing students [40]. In this study, the participants encouraged each other and worked as a team with their colleagues. Social support and self-efficacy can have positive effects on the nursing work environment for new nurses. Thus, it is important to establish a supportive atmosphere for medical staff and new nurses who are in charge of nursing COVID-19 patients in the future [41].

Fourth, in the last theme cluster “positive changes through patient care,” new nurses gained confidence after nursing COVID-19 patients and were more confident about their roles and responsibilities as nurses. When military medical staff are dispatched to unfamiliar environments, they often face unique situations in which they need to care for patients using limited medical equipment and supplies. Furthermore, military nurses are tasked with important duties that involve immediately conducting various tasks, in addition to general nursing, unit management, and participation in training [42] in national crisis situations. Thus, the work environment and conditions of new military nurses differ from those of nurses working in private medical institutions. Although these differences may be another stress factor for military nurses, the participants in this study felt proud, as they recognized their roles and planned patient-centered nursing in the isolation units. These findings are similar to those of previous studies. New nurses with experiences in nursing COVID-19 patients initially felt psychological and physical discomfort due to the fear of caring for those with a new infectious disease and working in an isolated environment. However, the new nurses overcame these hardships to develop professional values, which aided them in their job [19]. In addition, new nurses experienced fear and anxiety; however, they felt fulfilled and rewarded as they continued to nurse the patients [43]. These changes may be attributed to the increased adaptability of new nurses to work, and reduced stress and burnout [35].

As described, this study is meaningful in that it explored the experiences of new nurses caring for COVID-19 patients in military hospitals and investigated these experiences from the perspective of nurses during the unprecedented COVID-19 pandemic. In particular, the participants in our study were active-duty nursing officers serving in military hospitals, whose experiences are not well-known because of the specificity and conservative nature of the military. However, several limitations must be considered in the interpretation of this study’s findings. First, the participants of this study were new nurses from military hospitals, which are NDIUs for COVID-19 in South Korea. Thus, the different regional and environmental specificities may lead to different experiences in other cultures and conditions. Second, data were collected through telephone interviews due to the constraints caused by the COVID-19 pandemic. Although this study aimed to understand the emotions and thoughts of the participants from various angles and while data were carefully interpreted, non-verbal data could not be collected.

## 5. Conclusions

This study sought to obtain a holistic understanding of the vivid experiences of new nurses who were immediately deployed in a national medical crisis when caring for COVID-19 patients in NDIUs in military hospitals, using the phenomenological method. These findings provided insights into how new nurses perceived such experiences at military hospitals. The results indicated the need to prepare practical educational programs and policies for various infectious diseases to improve the nursing capacity of new nurses at military hospitals. This study is meaningful, as the basic data that were identified may be applied while formulating policies related to patients with infectious diseases, as well as for military nursing personnel caring for these patients. Moreover, this study may serve as the basis for mission training courses to prepare military medical personnel for national crisis situations requiring immediate responses. Based on the findings, further studies must analyze nursing competencies and factors affecting the care of COVID-19 patients and other general infectious diseases. In addition, comparative studies on the experience of nursing infectious disease patients among new nurses in military and private hospitals must be conducted.

## Figures and Tables

**Table 1 healthcare-10-00744-t001:** Examples of key interview questions.

Key Interview Questions
● Describe any challenges and useful experiences of caring for COVID-19 patients as a new nurse?
● As a new nurse, how did you care for COVID-19 patients?
● What changes have you experienced after caring for COVID-19 patients?
● What does the experience of caring for COVID-19 patients mean to you?

**Table 2 healthcare-10-00744-t002:** Examples of formulated meanings, themes, and theme clusters.

Formulated Meanings	Themes	Theme Clusters
Confused by lack of knowledge and experience Anxiety about infection Worry of infecting others and family members Distressed due to prejudice of being seen as a risk factor	Fear of a new circumstance	Burden of nursing in isolation units
Communication difficulties in non-face-to-face situations Patients and families who do not understand isolation care Protective equipment interfering with communication	Communication difficulties with isolated patients	

**Table 3 healthcare-10-00744-t003:** Themes and theme clusters of the study.

Theme Clusters	Themes
Burden of nursing in isolation units	Fear of a new circumstance
	Communication difficulties with isolated patients
Hardship of nursing critically ill patients	Nervous about caring for unfamiliar, critically ill patients
	Physically exhausted
	Psychological withdrawal
Efforts to perform nursing tasks	Studying hard to provide skilled nursing
	Searching for own know-how
	Showing comradeship and encouraging each other
Positive changes through patient care	Gaining confidence
	Striving for patient-centered care
	Thinking critically
	Feeling proud as a military nursing officer

## Data Availability

Not applicable.

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
