# Peer review of "Nursing Experience of New Nurses Caring for COVID-19 Patients in Military Hospitals: A Qualitative Study"

_healthcare, 2022, doi:10.3390/healthcare10040744_

Round 1

Reviewer 1 Report

The present manuscript does not presnet a clear paradigm to be challenged, and it is not clearly presented in its hypothesis and aims. What was the paradigm? What it is the new information generated to the public? What it is the novelty involved in the type of study? What was the rationale to use the statistical analysis? There are no details of the patients. 

To me, that it is the main flaw of the present manuscript, which it is very reduced in its study population, with a low number of cases. As it is, to me, it is a very prliminar study with no final conclussions that can be withdrawn from it.

Reviewer 2 Report

This is a well-written paper. To fully understand the research design, clarifications in the Methods sections are needed:

Some information regarding the interviewers should be included here. Was it a 1 on 1 interview in every case? Was it the same interviewer in every case, or were there different interviewers? Their credentials, occupation, gender, experiences, whether they had established relationships with participants…the personal characteristics of the research team are an important part of understanding participants’ responses. For instance, if participants knew that the interviewers had organizational power over them or were also military medical professionals, they may give different answers than if interviewers were entirely outside the participants’ professional sphere. The informed consent procedure includes this reflexivity (“The first author held a 139 senior position in the same institution as the participants, which risked limiting their voluntary participation; hence, the corresponding author who did not work in the same institution as the participants, and had no conflicting interests, explained the purpose and procedure of this study, the participants’ right to voluntarily participate and withdraw from the study, and the guarantee of anonymity and confidentiality through individual phone consultations, to confirm the intention for participation”) and it would be helpful to see it in regards to interviews as well.

Page 3, “Each participant was interviewed two to three times until sufficient data were collected.” Please include some mention here of how researchers defined “sufficient” data. Was it topic saturation or some other rubric?

What language were interviews conducted in? If Korean, please specify how translation was handled.

In Methods it says, “Due to the COVID-19 pandemic, data were collected face-to-face or over the phone, based on the participants' preference…The interviews were conducted in a quiet and comfortable seminar room in the hospital.” but then in page 11 in the Discussion a limitation is listed as, “data were collected through telephone interviews due to constraints caused by the COVID-19 pandemic.” Please clarify how many interviews were conducted via telephone and how many were face-to-face.
